# Prevalence and Phenotypic Antimicrobial Resistance among ESKAPE Bacteria and Enterobacterales Strains in Wild Birds

**DOI:** 10.3390/antibiotics11121825

**Published:** 2022-12-15

**Authors:** Tamara Pasqualina Russo, Adriano Minichino, Antonio Gargiulo, Lorena Varriale, Luca Borrelli, Antonino Pace, Antonio Santaniello, Marina Pompameo, Alessandro Fioretti, Ludovico Dipineto

**Affiliations:** 1Department of Veterinary Medicine and Animal Production, Federico II University of Naples, 80134 Naples, Italy; 2Marine Turtle Research Group, Department of Marine Animal Conservation and Public Engagement, Stazione Zoologica Anton Dohrn, 80055 Portici, Italy; 3ASL Napoli 1 Centro, Veterinary Hospital, 80145 Naples, Italy

**Keywords:** wild birds, ESKAPE bacteria, antimicrobial resistance, One Health, enterobacterales, multidrug-resistant

## Abstract

Antimicrobial resistance (AMR) is a current public health issue globally. To counter this phenomenon and prioritize AMR in the health sector, the World Health Organization (WHO) published a list of bacterial pathogens against which the development of new antimicrobial agents is urgently needed, designating the ESKAPE pathogens (i.e., *Enterococcus faecium*, *Staphylococcus aureus*, *Klebsiella pneumoniae*, *Acinetobacter baumannii*, *Pseudomonas aeruginosa*, and *Enterobacter* species) with a ‘priority status’. Moreover, the One Health High Level Expert Panel (OHHLEP) states that human health is closely linked to animal and environmental health, thus promoting a holistic One Health approach in order to be prepared to address possible emerging health threats from the human–animal–environment interface. Wild birds may host and spread pathogens, integrating the epidemiology of infectious diseases. The aim of this study was to examine the role of wild birds as a source of ESKAPE bacteria and other antibiotic-resistant enterobacterales. A total of fifty strains within the ESKAPE group were detected in 40/163 cloacal samples of examined birds (24.5%). Additionally, different strains of enterobacterales were detected in 88/163 cloacal samples (53.9%). Isolated strains exhibited antimicrobial resistance, including towards critically important antibiotics (e.g., third, fourth, fifth generation cephalosporins, fluoroquinolones) for human medicine. Our results confirm that wild birds are potential reservoirs of several pathogens and antimicrobial-resistant bacteria and that they could be involved in the dissemination of those bacteria across different environments, with resulting public health concerns.

## 1. Introduction

The phenomenon of antimicrobial resistance (AMR) is currently one of the major public health issues worldwide, with relevant clinical and economic implications. The problem of AMR is complex because it involves several aspects, ranging from the indiscriminate use of these drugs in human and veterinary medicine, animal husbandry, and agriculture, to the increased spread of antibiotic-resistant microorganisms, including those responsible for infections. AMR is a natural phenomenon: the development of resistance can be the result of spontaneous mutations or the acquisition of resistance genes through horizontal gene transfer, such as conjugation, transformation, and transduction [1,2,3]. The constant use of antimicrobials increases the selective pressure: susceptible bacteria are killed or inhibited, whereas bacteria that are naturally or intrinsically resistant, or that have acquired resistance traits, have a greater chance of surviving and multiplying [4]. This promotes the emergence, multiplication, and spread of resistant strains. Moreover, the emergence of bacteria simultaneously resistant to more than three classes of antibiotics (multidrug resistance as defined by Magiorakos et al. [5]) has reduced the possibility of effective treatments. In recent years, to counter this phenomenon, the World Health Organization (WHO) and the European Centre for Disease Prevention and Control (ECDC), have provided recommendations and proposed coordinated strategies and actions, recognizing AMR as a priority in the health sector [6].

In addition, in 2017, the WHO published a list of bacterial pathogens against which the development of new antimicrobial agents is urgently needed. Specifically, within this extensive list, the ESKAPE pathogens (*Enterococcus faecium*, *Staphylococcus aureus*, *Klebsiella pneumoniae*, *Acinetobacter baumannii*, *Pseudomonas aeruginosa*, and *Enterobacter* species) were designated with a ‘priority status’ (WHO) [7]. The term ‘ESKAPE’ is an acronym used to identify bacterial agents characterized by marked virulence and high antibiotic resistance. ESKAPE bacteria are responsible for nosocomial infections, are able to ‘escape’ the biocidal action of antibiotics [7,8], and are associated with an increased risk of mortality in humans [9]. Along with *Escherichia coli*, ESKAPE bacteria have caused the majority of life-threatening bacterial infections in healthcare facilities among critically ill and immunocompromised patients worldwide [10]. However, neither commensal *E. coli* nor ESKAPE bacteria are generally pathogenic [11], because most of them (i.e., *E. coli*, *S. aureus*, *K. pneumoniae*, *Enterobacter* spp., *Enterococcus* spp.) are commonly harbored in humans and animals [10]. Conversely, *Acinetobacter* spp. and *P. aeruginosa* prevail in soil and aquatic environments, but information on their prevalence in animals and their transmission from animals to humans is still scarce [10].

Human health is closely linked to animal and environmental health, so it is important to consider a holistic One Health approach in order to prevent and address possible emerging health threats from the human–animal–environment interface. Indeed, the One Health High Level Expert Panel (OHHLEP) states: “One Health is an integrated and unifying approach that aims to sustainably balance and optimize the health of people, animals, and ecosystems. […] It recognizes that the health of humans, domestic and wild animals, plants, and the broader environment (including ecosystems) are closely connected and interdependent” [12]. In light of the above, and also considering the worldwide spreading of the antibiotic resistance emergency, which attracts attention and interest from the scientific community, the aim of this study was to assess the presence and antibiotic resistance of ESKAPE bacteria and other enterobacterales in wild birds in the Campania region (southern Italy).

Wild animals are widely reported as reservoirs of pathogens and antibiotic resistance; birds, in particular, due to their ability to occupy different ecological niches and to adapt to many urban, suburban, and livestock environments, represent true sentinels. They reflect human activities and the impact of these activities on the environment, and can easily carry human and environmental bacteria. Resistant bacteria of human and veterinary origin are believed to be transmitted to wild birds through contaminated food or water [13,14,15,16]. In addition, birds, because of their ability to move and through the deposition of droppings, can play an important role as vectors in the environmental circulation and spread of zoonotic agents, antimicrobial-resistant bacteria, and resistance genes [17]. Numerous studies conducted worldwide have reported the presence of zoonotic bacteria and antibiotic-resistant strains in wild animals [18,19,20]. In particular, several studies have reported the presence of critically important antibiotic resistance genes (ARGs) in migratory birds [21,22,23,24], further highlighting that migratory birds may serve as a reservoir for the dissemination of antimicrobial resistance and of emerging AMR [22].

Specifically, in southern Italy, surveys conducted on several bird species have reported the presence of *Salmonella enterica* serovar *Infantis* in common swifts [25], thermotolerant *Campylobacter* in raptors [26], enteropathogenic bacteria in urban pigeons and gulls [27,28]. In addition, a study conducted by Foti et al., in 2017 [18], confirmed that migratory wild birds play an important role in the ecology and circulation of potential zoonotic pathogens, and detected resistance against two molecules belonging to the carbapenemes family, which are normally used only in human clinical practice as a last resort for the treatment of infections caused by antimicrobial-resistant bacteria. Antibiotic residues and bacteria carrying antibiotic resistance can be introduced into the environment due to the spread of medicated livestock effluent and urban effluent on agricultural land. Several studies have demonstrated a widespread prevalence of antibiotic-resistant enterobacteria in bird populations sympatric to human-inhabited areas and areas with high livestock densities [18]. However, the majority of the epidemiological studies on wildlife available in the literature are focused on single bacterial species, not considering the entire ESKAPE group. Therefore, our study is aimed at acquiring new data on the presence of bacteria belonging to the ESKAPE group and other enteric bacteria, from wild birds in southern Italy. Moreover, the phenotypic resistance of the isolated strains to different classes of antibiotics was evaluated in order to assess the levels of antimicrobial resistance in free-living birds not subjected to antibiotic treatment, which can be therefore considered as an environmental mirror.

## 2. Results

A total of fifty strains within the ESKAPE group were detected in 40/163 cloacal samples of examined birds (24.5%; 95% Confidence Interval [CI] = 18.2–32.0%), with *n* =18 *Pseudomonas aeruginosa* (36.0%), *n* = 9 *Enterococcus faecium* (18.0%) and *n* = 7 *Enterobacter aerogenes* (14.0%) as the most common species, followed by *n* = 6 *Staphyloccoccus aureus* (12.0%), *n* = 6 *Enterobacter cloacae* (10.0%), and *n* = 4 *Klebsiella pneumoniae* (8.0%). *Acinetobacter baumannii* was never isolated during the investigation. The majority of birds hosted either one (29/40, 72.5%) or two (9/40, 22.5%) species, one animal hosted three (1/40, 11.1%) species. Additionally, different strains of enterobacterales were detected in 88/163 cloacal samples (53.9% 95% CI 46.0–61.7%), with *Escherichia coli* (*n* = 55, 62.5%) as the most common species followed by the genera *Citrobacter* spp. (*n* = 18, 32.7%), *Serratia* spp. (*n* = 9, 16.3%), *Proteus* spp. (*n* = 4, 7.2%) and *Salmonella* spp. (*n* = 2, 3.6%), as reported in Figure 1.

The bacteriological results and prevalence related to each species are summarized in Appendix A Appendix A.

The results of the antibiograms with the Kirby–Bauer method were significantly variable. The strains of *P. aeruginosa* showed high percentages of susceptibility to the chosen set of antibiotics, with seven isolates (38.8%) exhibiting intermediate susceptibility to piperacillin. All *E. faecium* strains were susceptible to ciprofloxacin and chloramphenicol; one isolate (11.1%) was resistant only to penicillin, whereas the 44% (4/9) were simultaneously resistant to penicillin, erythromycin, doxycycline, and tetracycline. The strains of *S. aureus* showed high resistance to the tested antibiotics. In particular, 100% of the isolates were resistant to clindamycin and cefoxitin and 66.6% were also resistant to erythromycin, doxycycline, tetracycline, and trimethoprim-sulfamethoxazole. Among the enterobacterales isolates, the resistances were determined for a total of 105 strains, including *Klebsiella* spp. and *Enterobacter* spp. The highest resistance values were found against ampicillin (59.0%), including the natural resistance of some isolated species, followed by amoxicillin/clavulanic acid (38.0%), tetracycline (26.6%), piperacillin (25.7%), trimethoprim-sulfamethoxazole and ceftriaxone (21.9%). Among all the strains of enterobacterales, 10/105 (9.52%) were potential ESBL positives, as detected by the double disc diffusion test, which resulted in a significant increase (≥5 mm) of the inhibition zone for the combination disc in comparison to the inhibition zone for the cefpodoxime disc alone. Finally, *n* = 24 (22.8%) enterobacterales, *n* = 4 (2.89%) *S. aureus,* and *n* = 4 (2.89%) *Enterococcus* spp. here isolated, were categorized as multidrug-resistant (MDR) strains. All the details of the antibiogram results are reported in Appendix A Appendix A.

## 3. Discussion

In this study, 163 bird cloacal swabs from the Wildlife Rescue Center of the University of Naples Federico II (Italy) were collected and examined. We investigated the presence and the levels of antimicrobial resistance of bacteria belonging to the ESKAPE group and other enteric bacteria. Data available in the literature cover several countries and refer to numerous bird species belonging to different orders and families, often characterized by completely different origins, habits, feeding, population density, and distribution. This great variability results in varied data, not always allowing a linear comparison. A total of fifty ESKAPE bacteria was isolated from 40/163 birds from mixed infections, with *P. aeruginosa*, *E. faecium,* and *E. aerogenes* as the most common species, followed by *S. aureus*, *E. cloacae,* and *K. pneumoniae*.

The results collected during our survey are not surprising and confirm the role of birds as reservoirs of pathogens and antibiotic-resistant strains. All wild birds can, therefore, through their excretions, be sources of pathogenic bacteria and infectious agents for other animals both wild and domestic, as well as humans. Among the many wild bird species, long-range migratory can play an important role in the ecology, circulation, and dissemination of enteric human pathogens [18]. In rest areas, birds of different species meet, promoting the horizontal transmission of pathogens [29]. The majority of birds sampled in our study belong to bird species that occupy specific environmental niches, often abutting the urban environment. Birds of prey, in particular, being at the top of the food chain, are excellent ecological indicators. In fact, not surprisingly, the majority of multiresistant bacteria were recovered in birds of prey and in resident birds. Several studies have shown a wide spread of antibiotic-resistant enterobacteria in bird populations sympatric to areas inhabited by people and areas with a high density of livestock [30,31,32].

In the last decade, the presence of these bacterial agents from birds has been reported in several countries, and many authors have studied their characteristics and mechanisms related to the AMR phenomenon, describing the potential role of wild birds in harboring and spreading resistant bacteria as a potential hazard to human and animal health [19,33,34,35]. In 2019, Ahmed et al. [19] reported the presence of *E. coli*, *K. pneumoniae*, *K. oxytoca,* and *P. aeruginosa* from resident and migratory wild bird samples in Egypt with different prevalence rates. In particular, *P. aeruginosa* was detected, respectively, in 10.0% and in 18.3% of birds resident and migratory in line with our study where this bacterium species was the most represented of the ESKAPE group, with a total prevalence of 11.1% (95% CI 6.85–17.1%). In contrast, a recent study conducted in England by Rodrigues et al. [36], reported the presence of *Pseudomonas* spp. in fecal samples from wild birds with a prevalence of 20.9%, with *P. aeruginosa* detected in only one bird. Another finding that differs from our study concerns the antibiotic resistance profiles: *P. aeruginosa* strains tested in our study showed high percentages of susceptibility to the chosen set of antibiotics and only seven isolates (38.8%) exhibited intermediate susceptibility to piperacillin, whereas the previous authors reported that 87.5% of the strains were resistant to ciprofloxacin and 83.3% were resistant to cefepime.

High levels of antibiotic resistance genes were also detected in *Enterococcus* spp. and *P. aeruginosa* isolated from the feces of three common wild bird species (ducks, crows, and gulls) in Houston metropolitan areas [37], with 52.5% of *Enterococcus* spp. and 45.0% of *P. aeruginosa* containing both antimicrobial resistance and virulence genes. In our study, *E. faecium* was the second most represented species in the ESKAPE group, but with a generally low prevalence (5.5%), in contrast to a previous study conducted in Poland, where prevalence rates ranged from 55% in the feces of urban swallows to 81.5% in cloacal swabs of other wild birds [38]. In addition, the same authors reported that among the *E. faecium* strains tested, ten were resistant to vancomycin and teicoplanin. Another Polish study, conducted by Stępień-Pyśniak et al. [39], on different enterococci species isolated from wild bird species, reported that lincomycin resistance was the most common, followed by erythromycin, ciprofloxacin, and tetracycline, but no isolate was found resistant to vancomycin and chloramphenicol. These data are partially in line with our study since 44% of the strains we tested were found to be simultaneously resistant to penicillin, erythromycin, doxycycline, and tetracycline and all strains were susceptible to ciprofloxacin and chloramphenicol. However, information on vancomycin, teicoplanin, and lincomycin resistance is missing in our report because these antibiotics have not been tested.

*S. aureus* was detected in 6/163 (3.6%) of the birds we examined. This value is low when compared with other research conducted in different countries. Elsohaby et al. [40] isolated from migratory wild birds around Lake Al-Asfar *n* = 37 strains of *Staphylococcus* spp. and 9.5% of these were *S. aureus*. In addition, a study conducted on tracheal and cloacal swabs of nocturnal raptors reported a prevalence of *S. aureus* isolation of 20.9% [41]. A recent review article, which analyzed the literature regarding the determination of pooled prevalence of nasal, tracheal, and/or oral *S. aureus* (NTO) and methicillin-resistant *S. aureus* (MRSA), reports that the prevalence of *S. aureus* NTO carriage varies greatly with wildlife category and that the lowest pooled prevalence of *S. aureus* was obtained from wild birds (10.3%). However, it was found that *S. aureus* is not often the staphylococcal species associated with nasotracheal carriage in wild birds (excluding birds of prey) [42]. The *S. aureus* strains we isolated showed high resistance to the tested antibiotics. Specifically, 100% of the isolates were resistant to clindamycin and cefoxitin, and 66.6% were also resistant to erythromycin, doxycycline, tetracycline, and trimethoprim-sulfamethoxazole. In contrast, 50% of the strains tested by Silva et al. [41] were sensitive to all the tested antibiotics.

Excluding *A. baumannii*, which was not found during our investigation, *K. pneumoniae* was the least represented species of the ESKAPE group, accounting for 8.0% of the strains and only 2.4% (95% CI 0.78–6.56%) of the total isolates. *K. pneumoniae* is an important enterobacteriacea, considered to be one of the opportunistic pathogens causing a wide spectrum of diseases and showing increasing acquisition of antibiotic resistance [43]. In our study, multiresistant and potential ESBL-positive strains of *K. pneumoniae* were isolated. The prevalence we recorded was in line with the findings of other authors. In particular, Chiaverini et al. [44] in a very recent study conducted in central Italy isolated from *Pica Pica* a strain of *K. pneumoniae*, which showed resistance to several antimicrobials. Similarly, Foti et al. in 2017 [18] reported two strains of *K. pneumoniae*, isolated from cloacal samples of migratory birds in Italy, and exhibiting resistance to 14 molecules.

Our bacteriological analysis also led to the isolation of numerous strains of enterobacterales, with *Escherichia* (33.7%; 95% CI 26.6–41.6%) and *Citrobacter* spp. (11.0%; 95% CI 6.85–17.1%) among the most represented genera, followed by *Serratia* spp. (5.5%; 95% CI 2.71–10.5%), *Proteus* spp. (2.4%; 95% CI 0.78–6.56%) and *Salmonella* spp. (1.2% 95% CI 0.21–4.82%). The occurrence of such agents is quite common in wild birds [18,19,20]. Noteworthy are the levels of resistance found in our study: 59.0% of the tested strains (including *Klebsiella* species) were resistant to ampicillin, 38.0% to amoxicillin/clavulanic acid, 26.6% to tetracycline, 25.7% to piperacillin, and 21.9% to trimethoprim-sulfamethoxazole and ceftriaxone. In addition, 24/105 (22.8%) isolated strains were categorized as MDR strains. Our results about the levels of AMR are partially in agreement with other similar studies carried out in Italy [20] and in Poland [45]. Giacopello et al. [20], in their study on the antimicrobial susceptibility profiles of enterobacteriaceae members isolated from cloacal swabs of common European wild birds, reported frequently displayed resistance to trimethoprim/sulfamethoxazole, followed by streptomycin, amoxicillin/clavulanic acid, and ampicillin. Lower levels of antibiotic resistance were detected to tetracycline. The highest levels of antibiotic susceptibilities were ceftazidime, norfloxacin, cefotaxime, and ciprofloxacin. In contrast, our results show that tetracycline resistance was the third most recorded resistance along with ceftriaxone, cefpodoxime, ampicillin, azithromycin, and ciprofloxacin; lower levels of antibiotic resistance were detected to chloramphenicol and imipenem. In a survey conducted on *E. coli* strains isolated from the feces of 34 wild birds of different species in suburban areas of Poland [45], resistances to tetracycline (50%), ciprofloxacin (46.8%), gentamicin (34.3%) and ampicillin (28.1%) were the most frequently reported. In addition, as many as 31.2% of *E. coli* isolates exhibited a multidrug resistance phenotype. Differently, in our report, gentamicin resistance was recorded in 5.7% of all enterobacterales tested and only in two *E. coli* strains (3.63%) out of a total of 55 isolates. The presence of *Salmonella* spp. was detected in 2/163 birds examined (1.22%), specifically a *Columba livia* and a *Buteo buteo*. Our data differ from those of Millán et al. [46], who in 2004 reported isolation prevalence of 8.5% in Spanish birds. In contrast, a study conducted by Botti et al. in Italy [47], reported *Salmonella* isolation frequencies closer to ours, with *Salmonella* positivity of 2.2%, and reported that half of the isolations occurred in raptors. Therefore, as other authors hypothesized, the importance of wild birds in spreading *Salmonella* could be limited to those residing in areas that are highly contaminated by human waste or domestic animal manure [48,49]. In light of our results, synanthropic and wild birds are confirmed to be reservoirs of ESKAPE group bacteria and of antibiotic-resistant enterobacteria. In addition, 23.1% (95% CI 16.6–31.2%) of the total strains tested were MDR and 9.52% of enterobacterales resulted in potential ESBL positive. The relevant finding is that some isolates tested in this study showed resistance to penicillins, cephalosporins (cefoxitin and ceftriaxone), tetracyclines, or trimethoprim-sulfamethoxazole. According to the WHO classification, tetracyclines and sulfonamides were classified as highly important, aminoglycoside as critically important, and cephalosporins (3rd, 4th, and 5th generations) as the highest priority critically important antimicrobials for human medicine [50]. Several studies report the presence of critically important antibiotic-resistant genes in migratory birds. In particular, research conducted in Algeria by Loucif et al. [21] indicates the presence of NDM-5 and MCR-1 antibiotic resistance encoding genes in enterobacterales in *Ciconia ciconia*. In addition, Chen et al. [22], in 2019, reported the emergence of tet(X4)-encoding tigecycline resistance mechanism in *E. coli* strains from migratory birds. Metagenomic analysis by Cao et al. [51] showed that ARGs in bird species differ from one another in number, type, and abundance, but tetracycline resistance genes were the most prevalent group of ARGs in all species. Furthermore, bird species inhabiting the same environment tended to have similar composition of bacteria community and ARG composition. Compared with the microbiomes and resistomes in the environments, migratory birds harbored a lower phylogenetic diversity but had more ARGs.

In our manuscript, we did not obtain information on the genetic antimicrobial resistance of isolated bacteria because, due to a technical problem, molecular characterization of resistance genes could not be performed. Particularly for the *K*. *pneumoniae* strains, all of which were MDR and ESBL positive. Recent work, conducted in northern Italy by Thorpe et al. [52], analyzed 3482 genome sequences of *Klebsiella* spp. isolates on a large scale and found limited transmission between clinical and nonclinical settings, including communities, animals, and environments.

Therefore, further research on antibiotic resistance genes and mobile genetic elements of strains with whole-genome sequencing technology, comparative analysis between ESKAPE bacterial strains from wild birds and clinical strains are more significant for public health and the mechanism of spread of antibiotic resistance genes. Thus, these results clearly demonstrate that wildlife, including birds, represents an area of AMR surveillance studies that would benefit from further research due to the potential risk to humans, animals, and the environment [53,54].

### Limits

The study conducted, while showing important data on wildlife, has some limits, namely: Firstly, the large difference in the number and in the species of sampled birds does not allow extending the results over different families or orders. Moreover, we did not obtain information on the genetic antimicrobial resistance of isolated bacteria because, due to a technical problem, it was not possible to perform the molecular characterization of resistance genes.

## 4. Materials and Methods

### 4.1. Sampling

During the period between March and November 2019, a total of 163 wild birds were examined and recovered at the Wildlife Rescue Center of the University of Naples Federico II (Italy). All animals were found in different areas of the Campania region (Italy) and belonged to 32 different species. The birds were recognized by an ornithologist and classified according to the west Palearctic checklist of the Italian birdwatching association EBN [55]. Details and the number of animals examined are reported in Appendix A. Birds were sampled upon their arrival at the Center using sterile cotton-tipped swabs; from each animal, a cloacal swab was obtained by inserting a sterile swab impregnated with Phosphate Buffered Saline (PBS) (ThermoFisher, Oxoid, Milan, Italy) into the cloaca and gently rotating the tip against the mucosa. Swab samples were stored in sterile tubes containing PBS and transported under refrigeration conditions to the microbiological laboratory of the Department of Veterinary Medicine and Animal Production of the University of Naples Federico II for bacterial isolation. All samples were processed in the laboratory within 24 h of collection. The sampling procedures are part of the standard clinical examination and routine diagnostic investigations of recovered wild birds, in accordance with the current legislation—Directive 2010/63/EU [56].

### 4.2. Isolation and Identification of Bacterial

Samples were analyzed for detection of enterobacterales, *Pseudomonas aeruginosa*, *Staphylococcus aureus*, and enterococci, following the laboratory protocols based on ISO procedures (i.e., ISO 6579-1:2017; ISO 21528-2:2017; ISO 6888-1:1999/A2:2018), including the use of control organisms for quality check. Each collected swab was inoculated in sterile tubes containing 10 mL of Buffered Peptone Water (BPW) (ThermoFisher, Oxoid, Milan, Italy), vortexed for 1 min, and incubated at 37 °C for 24 h. After an enrichment in BPW, samples were streaked onto different media simultaneously, as follows: MacConkey Agar (ThermoFisher) for selective growth of Enterobacteriaceae; Cetrimide Agar (ThermoFisher) for the selective isolation and differentiation of *P. aeruginosa*; Baird-Parker Agar Base (ThermoFisher) for the isolation of *S.aureus*; Slanetz and Bartley Agar Base (ThermoFisher) for isolation of enterococci. All plates were incubated under aerobic conditions at 37 °C overnight and inspected for colonies identification after incubation. In addition, for all samples, 100 µL enriched BPW were also inoculated in sterile tubes containing 10 mL of Rappaport-Vassiliadis broth base (ThermoFisher) for selective enrichment of *Salmonella* species and incubated at 42 °C for 24 h and then for a further 24 h at 37 °C into Xylose-Lysine-Desoxycholate Agar (ThermoFisher).

The colonies isolated on the different agar were primarily identified on the basis of their morphology, lactose metabolism, Gram’s staining technique, oxidase test, coagulase test, pigment production, and standard biochemical (in particular tests motility, indole, lactose/glucose fermentation, methyl red, citrate, urease, hydrogen sulfide, and gas production). American Type Culture Collection (ATCC) standard reference strains were used to verify the condition of incubation and the performance of the culture media. Each isolate was then confirmed using the Analytical Profile Index system (API System, bioMérieux, Marcy-l'Étoile, France), and the identification at the species level was considered successful when reading provided at least “Very Good id.” (%id > 99.0 and T > 0.5).

### 4.3. Determination of the AMR

Antimicrobial resistance of bacterial isolates was determined by the disc diffusion method on Mueller–Hinton agar (ThermoFisher) according to the Clinical and Laboratory Standards Institute (CLSI) guidelines [57]. The following antimicrobials were used at the concentrations shown: piperacillin (PRL 100 μg), ampicillin (AMP 10 µg), penicillin (P 10 units), amoxicillin/clavulanic acid (AMC 30 μg), cefoxitin (FOX 30 μg), ceftazidime (CAZ 30 μg), ceftriaxone (CRO 30 μg), cefpodoxime (CPD 10 μg), imipenem (IMI 10 μg), colistin (CT 10 μg), μg), gentamicin (CN 10 μg), amikacin (AK 30 μg), streptomycin (S 10 µg), azithromycin (AZM 15 μg), erythromycin (E 15 μg), tetracycline (TE 30 μg), doxycycline (DO 30 μg), ciprofloxacin (CIP 5 μg), clindamycin (DA 2 μg), trimethoprim-sulfamethoxazole (SXT 1.25/23.75 μg), chloramphenicol (C 30 μg), and rifampicin (RD 5 μg) (Antimicrobial Susceptibility Discs, ThermoFisher). Isolates were classified as susceptible, intermediate, or resistant according to the interpretation of the zone diameter as recommended by CLSI [57]. Potential production of extended-spectrum beta-lactamase (ESBL), as indicated by resistance to ceftazidime and by an inhibitory effect of clavulanic acid, was confirmed using a double disc diffusion test (Cefpodoxime Combination Disc Kit, TermoFisher). A multidrug-resistant (MDR) strain was defined as a strain resistant to at least three different classes of antimicrobials [5]. A reference control organism was used to verify the quality and accuracy of the testing procedures of antimicrobials.

### 4.4. Data Analysis

All information and results collected were entered and ordered using Microsoft Excel spreadsheet software; the created data were checked for completeness, clarity, accuracy, and consistency and analyzed using Vassarstats statistical software.

## 5. Conclusions

Our results confirm that wild birds are reservoirs of pathogens and bacteria resistant to highly and critically important antimicrobials. Due to their ability to move and through the deposition of droppings, birds can play an important role as vectors in the environmental circulation and spread of zoonotic agents, antimicrobial-resistant bacteria, and resistance genes [17]. The presence of potential clinically relevant ESKAPE bacteria (i.e., *K. pneumoniae*), ESBL-producing bacteria, and MDR strains is not surprising; in fact, birds, due to their ability to occupy different ecological niches and adapt to many urban, suburban, and livestock environments, represent true environmental sentinels.

## Figures and Tables

**Figure 1 antibiotics-11-01825-f001:**
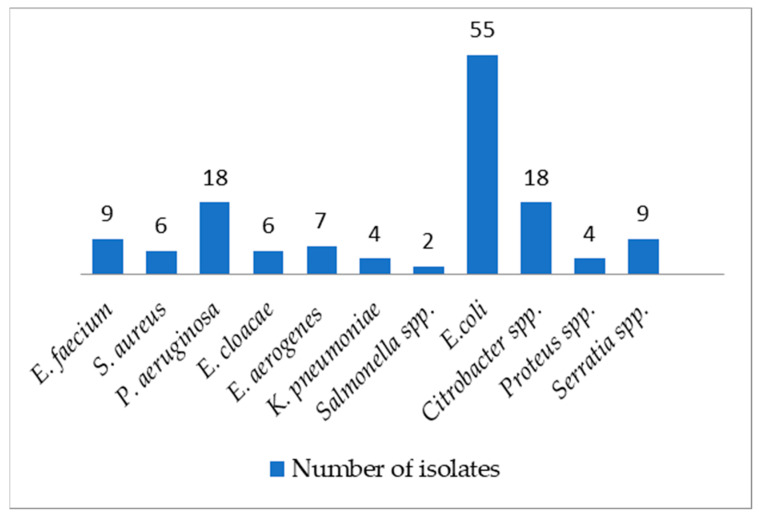
Number and species of isolates.

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
