# Peer review of "Prevalence and Phenotypic Antimicrobial Resistance among ESKAPE Bacteria and Enterobacterales Strains in Wild Birds"

_antibiotics, 2022, doi:10.3390/antibiotics11121825_

Round 1
Reviewer 1 Report
1. Has anyone studied the “Prevalence and phenotypic Antimicrobial Resistance among
ESKAPE bacteria and enterobacterales strains in wild birds” before? how this article is special than others (mention in abstract section), please explain the novelty of this manuscript in the abstract or introduction.
2. Pages must be numbered, and lines must be numbered consecutively throughout the manuscript. Also, the formats of references in this version are not consistent. The authors should recheck carefully the manuscript and make necessary revisions.
3. To meet the quality of the journal, editorial proofreading throughout the manuscript is suggested.
4. There are no tables or pictures in the article, so it is impossible to state your own views and research results.
Author Response
Dear Reviewer,
We were delighted to receive your review report and would like to thank you for the suggestions. We have now revised our manuscript in light of the comments put forth by you and the other reviewers. Below, please find our point-by-point response, indicating how and where in the manuscript changes have been made.
- Has anyone studied the “Prevalence and phenotypic Antimicrobial Resistance among ESKAPE bacteria and enterobacterales strains in wild birds” before? how this articleis special than others (mention in abstract section), please explain the novelty of this manuscript in the abstract or introduction.
RE: Thank you for your comment. We have added in the introduction section this sentence: “However, the majority of epidemiological studies on wildlife available in literature are focused on single bacterial species not considering the entire ESKAPE group. Therefore, our study is aimed at acquiring new data on the presence of bacteria belonging to the ESKAPE group and other enteric bacteria, from wild birds in southern Italy”.
- Pages must be numbered, and lines must be numbered consecutively throughout the manuscript. Also, the formats of references in this version are not consistent. The authors should recheck carefully the manuscript and make necessary revisions.
RE: Pages and lines were numbered accordingly. Manuscript was rechecked and improved.
- To meet the quality of the journal, editorial proofreading throughout the manuscript is suggested.
RE: Editorial proofreading was completed.
- There are no tables or pictures in the article, so it is impossible to state your own views and research results.
RE: All results are available in the supplementary tables. To make the information more direct, we have included a chart on the number and species of bacteria isolated.
Reviewer 2 Report
ABSTRACT
1) Please include the meaning of the acronym ESKAPE.
2) What are the vital antibiotics? Citing one or two examples would be enriching to the summary.
3) “pathogens and bacteria resistant,”… The sentence seems incomplete, resistant bacteria to what?
INTRODUCTION
4) page 2: Please include the reference that subsidizes the definition of multidrug resistance.
5) page 2: Please include the references related to the priority status in the health sector by ECDC and WHO.
6) page 2: Although the authors have put the term "collect,” the sentence does not seem accurate. Please rewrite it for clarity.
RESULTS
7) Please standardize in Supplementary Table 2 the presentation of the bacterial genus and species.
8) Please, I could not find the antimicrobial data concerning E. faecium, P. aeruginosa, and S. aureus in the supplementary materials.
DISCUSSION
9) Please put all scientific names in italics in this topic and where else is needed.
10) page 4: Silva's quote needs to be completed in the penultimate paragraph.
11) page 5: “with Salmonella positivity di 2.2% and reporting that half of the isolations occurred in raptors.” - please check that the sentence is correct.
12) I missed more discussion about the One Health approach. But mainly about an ecological perspective. Which species of birds migrate and are reservoirs of multiresistant bacteria? Is there a greater or lesser impact of spreading these microorganisms at the expense of these specific bird species? Which is the one that travels longer migratory distances and, therefore, could carry the pathogens to other habitats? Do any of these birds fly over near urban environments, etc.? Please discuss these and other topics that would be enriching to the manuscript.
MATERIALS AND METHODS
SAMPLING
13) About the taxonomic classification of birds, how was it made? By the authors of the article or by an expert on birds? Including the taxonomic manual/guidebook to which the listed species refer would enrich the work.
14) How/What method was used to retain the birds?
ISOLATION AND IDENTIFICATION OF BACTERIAL
15) Please include the ISO number.
16) “37° C” –> 37 ºC
17) Please standardize the presentation of units throughout the text; sometimes, they appear as 10mL... 10 mL.
DETERMINATION OF THE AMR
18) Please include the reference that subsidizes the definition of multidrug resistance.
GENERAL
19) Please include line numbers in the manuscript, as this makes it easier to mention comments on the text.
Author Response
Dear Reviewer,
We were delighted to receive your review report and would like to thank you for the suggestions. We have now revised our manuscript in light of the comments put forth by you and the other reviewers. Below, please find our point-by-point response, indicating how and where in the manuscript changes have been made.
ABSTRACT
1) Please include the meaning of the acronym ESKAPE.
RE: Thank you for your comment. The meaning of the acronym ESKAPE was added in the abstract.
2) What are the vital antibiotics? Citing one or two examples would be enriching to the summary.
RE: We would like to thank reviewer 2 for his/her suggestion. “Vitally” was changed in “critically” and some classes of antibiotics were added in brackets as an example.
3) “pathogens and bacteria resistant,”… The sentence seems incomplete, resistant bacteria to what?
RE: Sentence was modified in: ….“wild birds are potential reservoirs of several pathogens and antimicrobial resistant bacteria”
INTRODUCTION
4) page 2: Please include the reference that subsidizes the definition of multidrug resistance.
RE: Reference was added accordingly.
5) page 2: Please include the references related to the priority status in the health sector by ECDC and WHO.
RE: Reference was added accordingly.
6) page 2: Although the authors have put the term "collect,” the sentence does not seem accurate. Please rewrite it for clarity.
RE: We would like to thank the reviewer 2 for his/her suggestion. The sentence was rewritten as follows: They reflect human activities and the impact of these activities on the environment, and can easily carry human and environmental bacteria
RESULTS
7) Please standardize in Supplementary Table 2 the presentation of the bacterial genus and species.
RE: Table 2 was rechecked and improved.
8) Please, I could not find the antimicrobial data concerning E. faecium, P. aeruginosa, and S. aureus in the supplementary materials.
RE: The antimicrobial data concerning E. faecium, P. aeruginosa, and S. aureus have been added to the supplementary table.
DISCUSSION
9) Please put all scientific names in italics in this topic and where else is needed.
RE: Manuscript was rechecked and improved.
10) page 4: Silva's quote needs to be completed in the penultimate paragraph.
RE: reference was corrected accordingly.
11) page 5: “with Salmonella positivity di 2.2% and reporting that half of the isolations occurred in raptors.” - please check that the sentence is correct.
RE: Reported typos were corrected.
12) I missed more discussion about the One Health approach. But mainly about an ecological perspective. Which species of birds migrate and are reservoirs? Is there a greater or lesser impact of spreading these microorganisms at the expense of these specific bird species? Which is the one that travels longer migratory distances and, therefore, could carry the pathogens to other habitats? Do any of these birds fly over near urban environments, etc.? Please discuss these and other topics that would be enriching to the manuscript.
RE: Discussion was rechecked and improved.
MATERIALS AND METHODS
SAMPLING
13) About the taxonomic classification of birds, how was it made? By the authors of the article or by an expert on birds? Including the taxonomic manual/guidebook to which the listed species refer would enrich the work.
RE: The birds were recognized by an ornithologist and classified according to the west paleartic checklist of the italian birdwatching association EBN, respecting the taxonomic classification chosen by the International Ornithological Committee (IOC). Reference was added accordingly.
14) How/What method was used to retain the birds?
RE: Specific handling techniques for each bird species performed by veterinary specialist.
ISOLATION AND IDENTIFICATION OF BACTERIAL
15) Please include the ISO number.
RE: ISO number was added accordingly.
16) “37° C” –> 37 ºC
RE: Typos were corrected
17) Please standardize the presentation of units throughout the text; sometimes, they appear as 10mL... 10 mL.
RE: Units have been standardized.
DETERMINATION OF THE AMR
18) Please include the reference that subsidizes the definition of multidrug resistance.
RE: Reference was added accordingly.
GENERAL
19) Please include line numbers in the manuscript, as this makes it easier to mention comments on the text.
RE: Lines were numbered accordingly
Reviewer 3 Report
The manuscript by Tamara Pasqualina Russo and colleagues entitled “Prevalence and phenotypic Antimicrobial Resistance among ESKAPE bacteria and Enterobacterales strains in wild birds” the role of wild birds as a reservoir of ESKAPE bacteria and other Enterobacterales antibiotic-resistant strains. They found that 24.5% of the 163 birds carried 50 ESKAPE group bacteria, and enterobacterales were detected in 88/163 cloacal samples (53.9%).
Overall, this is a valuable piece on this important matter, and would inspire many follow up studies thereafter. However, the following minor points requires authors' attention.
Minor:
Introduction and Discussion section
The authors miss several references: There is a large body of literature on the diversity and abundance of highly and critically important antibiotic resistant microbes and antibiotic resistance genes in migratory bird, for example: https://doi.org/10.1016/j.scitotenv.2021.152861, https://doi.org/10.1186/s40168-019-0781-8, https://doi.org/10.1016/S2542-5196(19)30083-X, https://doi.org/10.1080/22221751.2019.1653795, https://doi.org/10.1016/j.envint.2018.05.039. It is concerning that none of these papers was cited.
In the manuscript, the authors also acknowledges the shortcomings that they did not obtain information on genetic antimicrobial resistance of isolating bacteria because, due to a technical problem, it was not possible to perform the molecular characterization of resistance genes.
Moreover, a recent paper through large-scale genomic snapshot of 3,482 genome sequences of Klebsiella spp. isolates in Northern Italy and found that limited transmission between clinical and non-clinical settings including community, animal and environmental settings. (Reference: https://doi.org/10.1038/s41564-022-01263-0)
Therefore, further research on antibiotic resistance genes and mobile genetic elements of strains whole genome sequencing technology, comparative analysis between ESKAPE bacteria strains from wild birds and clinical strains is more significant for public health and the dissemination mechanism of antibiotic resistance genes.
Author Response
Dear Reviewer,
We were delighted to receive your review report and would like to thank you for the suggestions. We have now revised our manuscript in light of the comments put forth by you and the other reviewers. Below, please find our point-by-point response, indicating how and where in the manuscript changes have been made.
The manuscript by Tamara Pasqualina Russo and colleagues entitled “Prevalence and phenotypic Antimicrobial Resistance among ESKAPE bacteria and Enterobacterales strains in wild birds” the role of wild birds as a reservoir of ESKAPE bacteria and other Enterobacterales antibiotic-resistant strains. They found that 24.5% of the 163 birds carried 50 ESKAPE group bacteria, and enterobacterales were detected in 88/163 cloacal samples (53.9%). Overall, this is a valuable piece on this important matter, and would inspire many follow up studies thereafter. However, the following minor points requires authors' attention.
RE: We would like to thank the Reviewer for her/his kind comments about our manuscript.
Minor:
Introduction and Discussion section
The authors miss several references: There is a large body of literature on the diversity and abundance of highly and critically important antibiotic resistant microbes and antibiotic resistance genes in migratory bird, for example: https://doi.org/10.1016/j.scitotenv.2021.152861, https://doi.org/10.1186/s40168-019-0781-8, https://doi.org/10.1016/S2542-5196(19)30083-X, https://doi.org/10.1080/22221751.2019.1653795, https://doi.org/10.1016/j.envint.2018.05.039. It is concerning that none of these papers was cited.
RE: References were added accordingly.
In the manuscript, the authors also acknowledges the shortcomings that they did not obtain information on genetic antimicrobial resistance of isolating bacteria because, due to a technical problem, it was not possible to perform the molecular characterization of resistance genes.
Moreover, a recent paper through large-scale genomic snapshot of 3,482 genome sequences of Klebsiella spp. isolates in Northern Italy and found that limited transmission between clinical and non-clinical settings including community, animal and environmental settings. (Reference: https://doi.org/10.1038/s41564-022-01263-0)
Therefore, further research on antibiotic resistance genes and mobile genetic elements of strains whole genome sequencing technology, comparative analysis between ESKAPE bacteria strains from wild birds and clinical strains is more significant for public health and the dissemination mechanism of antibiotic resistance genes.
RE: We would like to thank the reviewer 2 for his/her suggestion. Discussion was rechecked and improved
Reviewer 4 Report
Antimicrobial resistance (AMR) is now affecting public health all over the world. In this research, the authors looked at the function of ESKAPE bacteria and other enterobacterales antibiotic-resistant strains as sources from wild birds. They confirm that wild birds are likely to harbor diseases and germs that are resistant to treatment, and they also suggest that they may be implicated in the spread of these bacteria throughout various settings, raising issues for the general public's health. The results are intriguing, and I have a few minor comments for improving them.
Comments:
· In determination of AMR section line 4 write as AMP 10 ug.
· In same section line 8 write short form of erythromycin in the bracket
· In data analysis section line 3 remove extra analyzed using word
Author Response
Dear Reviewer,
We were delighted to receive your review report and would like to thank you for the suggestions. We have now revised our manuscript in light of the comments put forth by you and the other reviewers. Below, please find our point-by-point response, indicating how and where in the manuscript changes have been made.
Antimicrobial resistance (AMR) is now affecting public health all over the world. In this research, the authors looked at the function of ESKAPE bacteria and other enterobacterales antibiotic-resistant strains as sources from wild birds. They confirm that wild birds are likely to harbor diseases and germs that are resistant to treatment, and they also suggest that they may be implicated in the spread of these bacteria throughout various settings, raising issues for the general public's health. The results are intriguing, and I have a few minor comments for improving them.
Comments:
- In determination of AMR section line 4 write as AMP 10 ug.
- In same section line 8 write short form of erythromycin in the bracket
- In data analysis section line 3 remove extra analyzed using word
RE: We would like to thank the Reviewer for her/his kind comments about our manuscript.
Reported typos were corrected.
Round 2
Reviewer 1 Report
Manuscript entitled “Prevalence and phenotypic Antimicrobial Resistance among ESKAPE bacteria and enterobacterales strains in wild birds” has adequately addressed the previous concerns and the re-submission has been greatly improved, it’s comprehensive to convince me for accepting the manuscript.
Reviewer 2 Report
Lines 297 to 299, it is unnecessary to justify not performing molecular assays. To me, this period should be removed.
Otherwise, the authors answered all the questions and made the suggested improvements, so I recommend publishing the paper.